Effects of virtual reality intervention dosage on gait performance in Parkinson’s disease: a systematic review and meta-analysis

Yu Shanan 1
Zhu Yu 2
Yang Yanfei Tyjysyyf@163.com 3
1 College of Sport, Shanghai University of Electric Power , Shanghai , China
2 School of Physical Education, Shanghai University of Sport , Shanghai , China
3 Guangxi Health Science College , Guangxi , China
Anson Lesley
Electronic publication date: 2025 Nov 21
Publication date: 2025
Volume: 13
Electronic Location ID: e20320
Received 2025 Apr 28; Accepted 2025 Oct 10
Copyright: ©2025 Yu et al.
Copyright year: 2025
Copyright holder: Yu et al.
License: This is an open access article distributed under the terms of the Creative Commons Attribution License, which permits unrestricted use, distribution, reproduction and adaptation in any medium and for any purpose provided that it is properly attributed. For attribution, the original author(s), title, publication source (PeerJ) and either DOI or URL of the article must be cited.
License URL: https://creativecommons.org/licenses/by/4.0/

Keywords: Parkinson’s disease, Virtual reality, Gait rehabilitation, Motor function, Systematic review

Funding: The authors received no funding for this work.

==============================
Objective

Gait impairment is a prevalent and disabling feature of Parkinson’s disease (PD) that is often insufficiently improved by conventional rehabilitation approaches. Virtual reality (VR)-based training has emerged as a novel therapeutic strategy; however, the overall efficacy of VR interventions on gait outcomes in PD remains inconclusive. This systematic review and meta-analysis aimed to (1) quantify the effects of VR-based rehabilitation on gait performance in individuals with PD, and (2) investigate whether treatment outcomes are moderated by intervention dosage parameters-such as training frequency, session duration, and total intervention period-as well as patient-related factors like disease duration and age.

Methods

Seven databases (Embase, Web of Science, PubMed, Cochrane Library, Wanfang Data, VIP, and CNKI) were searched from inception to December 2024. Randomized controlled trials (RCTs) investigating VR training for gait in PD were independently screened by two reviewers. Study quality was assessed using the PEDro scale. Meta-analyses were conducted with RevMan 5.4.1, and publication bias was examined using Stata 17.0. Effect sizes were calculated using standardized mean differences with 95% confidence intervals. The certainty of evidence was graded using the GRADE approach.

Results

A total of 13 RCTs, involving 541 patients with Parkinson’s disease, met the inclusion criteria. VR-based interventions demonstrated significant effects in improving composite gait function (SMD = 0.56; 95% CI [0.36–0.77]; P < 0.00001), indicating a moderate and clinically meaningful benefit. Gait function was evaluated using a range of clinical scales and spatiotemporal parameters, encompassing multiple dimensions such as dynamic stability, walking efficiency, and functional mobility. Subgroup analyses revealed greater improvements among patients with disease duration ≤8 years (SMD = 0.65), shorter intervention periods (≤4 weeks, SMD = 0.86), and shorter session durations (≤30 minutes, SMD = 0.83). The intervention effects were generally consistent across different age groups (SMD = 0.48–0.74). The average PEDro score was 6.77, indicating moderate to high methodological quality, although allocation concealment and blinding were frequently absent. No significant publication bias was detected, and the overall certainty of evidence was rated as high.

Conclusion

VR-based training yields statistically robust and clinically relevant improvements in gait among individuals with PD. These benefits are moderated by disease stage and intervention parameters, supporting the integration of VR into personalized, early-phase rehabilitation strategies.

Introduction

Parkinson’s disease (PD) is a chronic and progressive neurodegenerative disorder marked by both motor and non-motor symptoms (Kalia & Lang, 2015), affecting approximately 0.3% of the general population and up to 2% of individuals over the age of 60 (Tysnes & Storstein, 2017). Among motor impairments, postural instability and gait disturbance (PIGD) are particularly disabling, especially in the mid-to-late stages of the disease (Honglin, Bao & Bin, 2018). A key manifestation of PIGD is freezing of gait (FOG)-a sudden, episodic inability to start or maintain walking-which affects a significant proportion of patients and is a major contributor to falls, injuries, and loss of independence (Wang, Zhu & Wang, 2024). These gait-related motor deficits are not only critical predictors of physical decline and reduced quality of life, but also impose substantial burdens on caregivers and healthcare systems, despite their clinical importance, gait disturbances in PD are often resistant to pharmacological treatment, highlighting the urgent need for effective, targeted rehabilitation approaches (Cai, Lin & Jiao, 2018).

Conventional rehabilitation approaches such as physical therapy and treadmill training have shown limited effectiveness in addressing the multifactorial mechanisms underlying gait dysfunction in PD. In recent years, virtual reality (VR)—based interventions have emerged as an innovative therapeutic option (Rong, Zhang & Liu, 2015). VR enables the creation of immersive, interactive environments that simulate real-life motor tasks, thereby engaging motor planning, visuospatial coordination, and sensorimotor integration (Calabrò et al., 2017). Electroencephalography (EEG)-based studies suggest that VR may activate brain networks involved in motor learning and cognitive control, including the mirror neuron system (Brunet-Gouet et al., 2016). Varying levels of immersion—non-immersive, semi-immersive, and fully immersive-allow for flexible tailoring of interventions (Ruidan, Peiling & Weijun, 2018). Among these, non-immersive VR has demonstrated potential in improving balance, gait, and activities of daily living in individuals with PD, while also being cost-effective and scalable (Jiang & Ren, 2013; Killane et al., 2015; Harris et al., 2015; Lin, Chen & Jiang, 2016).

Although several clinical trials and reviews have reported beneficial effects of VR-based training in PD, significant heterogeneity remains regarding intervention timing, session duration, and overall program length. The dose–response relationship and the impact of patient characteristics (e.g., age or disease stage) on treatment outcomes have yet to be clearly established (Harris et al., 2015; Laver et al., 2011). Moreover, it is unclear whether shorter or longer training regimens offer optimal benefits, and whether intervention efficacy varies by exposure frequency or duration per session.

In light of these uncertainties, this study aims to systematically evaluate the effectiveness of VR-based training on gait function in individuals with PD. Specifically, we seek to (1) determine the overall effect size of VR interventions on gait outcomes; (2) examine whether intervention efficacy is moderated by disease duration or patient age; and (3) explore the dose–response relationship in terms of intervention period, session duration, and training frequency. By synthesizing available evidence, this study seeks to offer clinically relevant guidance for optimizing VR-based rehabilitation protocols in Parkinson’s disease.

Data and Methods

This study adhered to the Meta-analysis PRISMA authoring guidelines (Page et al., 2021) for the selection and utilization of research methods and has been registered in the Prospective International Registry of Systematic Reviews and Meta-Analyses (PROSPERO) (registration number: CRD42024592834).

Research framework

This study adopts the framework of the International Classification of Functioning, Disability and Health (ICF) (Zhuoying, Lun & Di, 2020) to systematically examine how patient characteristics (e.g., age and disease duration) and VR intervention parameters (e.g., total duration, session length, and training frequency) influence improvements in lower-limb gait performance in individuals with Parkinson’s disease. The potential nonlinear dose–response relationship is also explored. The PICO structure of this systematic review is presented in Table 1.

Table 1 PICO architecture for virtual reality training interventions for gait in Parkinson’s disease patients.

Component	Description	
Population (P)	Individuals diagnosed with Parkinson’s disease (PD), with a mean age of ≥50 years across included randomized controlled trials (RCTs).	
Intervention (I)	Virtual reality (VR)-based training programs targeting gait improvement, implemented in various settings (e.g., clinical labs, rehabilitation centers, home-based environments) and delivered by qualified personnel (e.g., rehabilitation therapists). Interventions varied in duration (e.g., ≤4 weeks vs. >4 weeks), session length (e.g., ≤30 min vs. >30 min), and training frequency (e.g., 1–3 times/week vs.≥4 times/week).	
Comparison (C)	1. VR training group vs. non-VR or conventional training control group 2. Comparisons across VR intervention subgroups with varying dosage (duration, session time, frequency) 3. Subgroup comparisons based on patient age or disease duration	
Outcome (O)	Gait-related outcomes measured by validated clinical scales and spatiotemporal gait parameters, aligned with ICF domains: Gait-specific outcomes in individuals with Parkinson’s disease, including stride length, gait speed, cadence, balance, and dynamic stability, assessed using validated clinical rating scales and motion analysis parameters.	

Search strategy

The literature search was independently conducted by S.Y. and Y.Z. They systematically retrieved randomized controlled trials (RCTs) investigating VR training interventions for gait improvement in patients with Parkinson’s disease across seven databases: Embase, Web of Science, PubMed, The Cochrane Library, Wanfang Data, VIP, and China National Knowledge Infrastructure (CNKI), covering publications from database inception to August 2024. Manual searches of reference lists from relevant articles were also performed to ensure completeness. Any disagreements during the search process were resolved through discussion and consultation with a third referee, Y.Y. Detailed search strategies are presented in Table 2.

Table 2 Literature search strategy.

Comprehensive database	Search step	
PubMed and The Cochrane Library search strategies	#1(“Virtual Reality”[Mesh]) OR (‘Virtual Reality’[Title/Abstract] OR ‘Reality, Virtual’[Title/Abstract] OR ‘Virtual Reality, Instructional’[Title/Abstract] OR ‘Instructional Virtual Realities’[Title/Abstract] OR ‘Instructional Virtual Reality’[Title/Abstract] OR ‘Realities, Instructional Virtual’[Title/Abstract])
#2 (“Parkinson Disease, Secondary”[Mesh]) OR (‘Parkinson’[Title/Abstract] OR ‘Secondary Parkinson Disease’[Title/Abstract] OR‘Parkinson Disease, Symptomatic’[Title/Abstract] OR ‘Parkinsonism, Secondary’[Title/Abstract] OR ‘Parkinsonism, Symptomatic’[Title/Abstract] OR ‘Symptomatic Parkinsonism’[Title/Abstract] OR ‘Secondary Parkinson’[Title/Abstract] )
#3 (“Gait Apraxia”[Mesh]) OR (‘Gait’[Title/Abstract] OR ‘Apraxia, Gait’[Title/Abstract] OR ‘Apraxias, Gait’[Title/Abstract] OR ‘Gait Apraxias’[Title/Abstract] OR ‘Apraxia of Gait’[Title/Abstract] OR ‘Dyspraxia of Gait’[Title/Abstract] OR ‘Gait Dyspraxia’[Title/Abstract] OR ‘Gait Dyspraxias’[Title/Abstract] )
#4 (Randomized controlled trial [Publication Type] OR “Randomized” [Title/Abstract] OR “controlled” [Title/Abstract] OR “Trial” [Title/Abstract])
#5 #1 AND #2 AND #3 AND #4	
Web of Science Search Strategies	#1 TS=(‘Parkinson’ OR ‘Secondary Parkinson Disease’ OR ‘Parkinson Disease, Symptomatic’ OR ’Parkinsonism, Secondary’OR ‘Parkinsonism, Symptomatic’ OR ’Symptomatic Parkinsonism’ OR ‘Secondary Parkinson’ )
#2 TS=( ‘Virtual Reality’ OR ‘Reality, Virtual’ OR ‘Virtual Reality, Instructional’ OR ‘Instructional Virtual Realities’ OR ‘Instructional Virtual Reality’ OR ‘Realities, Instructional Virtual’)
#3 TS=(‘Gait’ OR ‘Apraxia, Gait’ OR ‘Apraxias, Gait’ OR ‘Gait Apraxias’ OR ‘Apraxia of Gait’ OR ‘Dyspraxia of Gait’ OR ‘Gait Dyspraxia’ OR ‘Gait Dyspraxias’)
#4 TS=(“Randomized controlled trial” OR “Randomized” OR “Controlled” OR “Trial”)
#5 #1 AND #2 AND #3 AND #4	
Embase Search Strategy	#1‘Parkinson’[exp] OR ‘Secondary Parkinson Disease’ [ab,ti] OR ’Parkinson Disease, Symptomatic’[ab,ti] OR ’Parkinsonism, Secondary’[ab,ti] OR ‘Parkinsonism, Symptomatic’ [ab,ti] OR ’Symptomatic Parkinsonism’[ab,ti] OR ‘Secondary Parkinson’ [ab,ti]
#2 ”Virtual Reality”[exp] OR ‘Virtual Reality’[ab,ti] OR ‘Reality, Virtual’[ab,ti] OR ‘Virtual Reality, Instructional’[ab,ti] OR ‘Instructional Virtual Realities’[ab,ti] OR ‘Instructional Virtual Reality’[ab,ti] OR ‘Realities, Instructional Virtual’[ab,ti]
#3 ”Gait Apraxia”[ab,ti] OR ‘Gait’[ab,ti] OR ‘Apraxia, Gait’[ab,ti] OR ‘Apraxias, Gait’[ab,ti] OR ‘Gait Apraxias’[ab,ti] OR ‘Apraxia of Gait’[ab,ti] OR ‘Dyspraxia of Gait’[ab,ti] OR ‘Gait Dyspraxia’[ab,ti] OR ‘Gait Dyspraxias’ [ab,ti]
#4 “Randomized controlled trial” [exp] OR “Randomized” [ab,ti] OR “Controlled” [ab,ti] OR “Trial” [ab,ti]
#5 #1 AND #2 AND #3 AND #4	
China Knowledge Network Search Strategy	Topic = (Parkinson’s + Parkinson’s disease + Parkinson’s disease patients + ‘Parkinson’s disease (pd)’ + Parkinson’s patients) AND Topic = (virtual reality + virtual reality technology + virtual reality systems + applications of virtual reality technology + ‘virtual reality (vr)’) AND Topic = (gait + gait analysis + gait anomalies + gait characterisation + gait training)	
Wanfang, Wipu search strategy	Topic = (Parkinson’s OR Parkinson’s disease OR Parkinson’s disease patients OR Parkinson’s disease (pd) OR Parkinson’s disease (pd)) AND Topic = (Virtual Reality OR Virtual Reality technology OR Virtual Reality technology applications) AND Topic = (Gait OR Gait training)	

Literature inclusion and exclusion criteria

Inclusion criteria

Studies were considered eligible for inclusion if they met the following criteria:

Population: Adults who were clinically diagnosed with Parkinson’s disease and exhibited moderate motor impairment, corresponding to Hoehn and Yahr stages 1 to 4 (Dockx et al., 2016; Rodríguez-Mansilla et al., 2023), and were between 60 and 80 years of age (De Lau & Breteler, 2006; Schrempf et al., 2014; Hughes et al., 1992).

Intervention: The experimental group received VR-based training, either as a standalone intervention or in combination with conventional rehabilitation programs, provided that no additional non-rehabilitative therapies (e.g., pharmacological agents, acupuncture) were involved. The inclusion of studies combining VR with standard rehabilitation was grounded in ethical and clinical considerations. In real-world Parkinson’s disease management, withholding conventional rehabilitation is generally not feasible due to safety concerns and standard-of-care obligations. Therefore, studies were included if VR was administered as an adjunct to routine rehabilitation, aligning with typical clinical practice and enhancing the external validity of the findings. Given the limited number of eligible studies and considerable variation in VR protocols and operational procedures, we did not further classify VR modalities (e.g., immersive vs. non-immersive, game-based vs. training modules). This limitation is discussed in the relevant section as a potential source of heterogeneity.

Comparison: The control group received conventional rehabilitation without any VR component.

Outcomes: The primary outcome was gait function, assessed using standardized clinical or quantitative measures. Specific parameters included gait stability and balance (measured by the Dynamic Gait Index (DGI), Functional Gait Assessment (FGA), and Timed “Up and Go” test (TUG); walking speed and endurance (10-Meter Walk Test (10MWT) and 6-Minute Walk Test (6MWT); spatiotemporal metrics (step frequency (m/s) and step length (cm)); global mobility and balance (Performance-Oriented Mobility Assessment (POMA total score)); motor symptom severity related to gait (Unified Parkinson’s Disease Rating Scale Part III-gait subscore (UPDRS-III gait)); and gait symmetry (left–right step length asymmetry).

Exclusion criteria

Studies were excluded if they met any of the following criteria: (1) incomplete data or lack of extractable quantitative outcomes; (2) use of VR in combination with pharmacological interventions or non-rehabilitative therapies, including but not limited to acupuncture, traditional Chinese medicine, electrical or magnetic stimulation, cognitive behavioral therapy, music therapy, aromatherapy, dietary supplementation, or psychotherapy. Importantly, studies combining VR with conventional rehabilitative approaches-such as physical therapy or gait training-were retained. Additional exclusion criteria included: (3) inconsistencies between the described intervention and the reported outcome measures; and (4) unavailability of the full text.

Literature screening and data extraction

All identified records were imported into EndNote for duplicate removal. Two reviewers (S.Y. and Y.Z.) independently screened titles, abstracts, and full texts against the eligibility criteria. Data were independently extracted and entered into RevMan 5.4.1, followed by cross-checking for consistency. Discrepancies were resolved through discussion with a third reviewer (Y.Y.) until consensus was reached.

Extracted data included the following information: First author and year of publication; Country of origin; Participant characteristics (age, sex, disease duration); Details of the VR intervention protocol; Reported outcome measures.

Quality assessment

The quality of the literature was assessed using the PEDro scale to evaluate the methodological quality of the included literature (Ludyga et al., 2020). The methodological quality of the included studies was evaluated using the 10-item Physiotherapy Evidence Database (PEDro) scale, which assesses the following criteria: random allocation, allocation concealment, baseline comparability, blinding of participants, blinding of therapists, blinding of outcome assessors, follow-up rate greater than 85%, intention-to-treat analysis, between-group statistical comparisons, and reporting of point estimates with variability measures. Each criterion was scored as 1 point if satisfied and 0 if not, yielding a total score ranging from 0 to 10. Studies were classified as low quality (scores < 4), moderate quality (4–5), good quality (6–8), or high quality (9–10). Only studies rated as moderate quality or above were included in this review.

Additionally, the certainty of evidence for each outcome was appraised using the GRADE (Grading of Recommendations Assessment, Development and Evaluation) framework via the GRADEpro tool. This system categorizes evidence into four levels: high, moderate, low, and very low. The assessment was independently conducted by two reviewers (S.Y. and Y.Z.). In cases of disagreement, a third reviewer (Y.Y.) was consulted, and discrepancies were resolved through discussion to reach consensus.

Data processing

All statistical analyses were conducted using RevMan 5.4.1. For each outcome, the mean, standard deviation, and sample size before and after intervention were extracted. As all outcome indicators were continuous variables, effect sizes were calculated using either the mean difference (MD) or standardized mean difference (SMD), depending on measurement consistency. Specifically, MD was used when studies employed the same instruments and units; SMD was applied when measurement tools or scales differed across studies to ensure comparability. Heterogeneity was assessed using Cochran’s Q test (P-value) and the I2 statistic. A P-value <0.05 combined with I2 > 50% was interpreted as significant heterogeneity, in which case a DerSimonian-Laird random-effects model was applied. If heterogeneity was not significant (P ≥ 0.05 and I2 ≤ 50%), a fixed-effects model was used. All effect estimates were presented with 95% confidence intervals (CIs) to indicate the precision and reliability of the pooled results. To evaluate the robustness of the findings, sensitivity analyses were performed using a leave-one-out method, whereby each study was removed in turn to examine its influence on the overall results. Publication bias for the primary outcomes was assessed using funnel plots and Egger’s regression test, both implemented in Stata 17.0.

Figure 1 Literature screening process.

Results

Results of literature search

A total of 953 records were identified through database searches and other sources. After removing duplicates (n = 920), 33 records remained for screening. Of these, 920 were screened by title and abstract, and 855 were excluded. Sixty-five full-text articles were then assessed for eligibility. Among them, 52 were excluded for reasons including full text unavailable (n = 7), unextractable outcomes (n = 10), irrelevant outcomes (n = 26), ineligible interventions (n = 4), conference abstracts (n = 2), and non-randomized designs (n = 3). Finally, 13 RCTs were included in both qualitative and quantitative synthesis. The selection process is illustrated in Fig. 1.

Basic information about the included literature

A total of 13 studies published between 2015 and 2023 were included in this review, encompassing 541 individuals diagnosed with PD (Santos et al., 2019; Pullia et al., 2023; Liao, Yang & Wu, 2015). Of these, 271 participants were assigned to the experimental group and 270 to the control group. In all included studies, the experimental group received VR-based interventions, whereas the control group underwent either conventional rehabilitation or received no specific intervention (i.e., blank control). Each study provided detailed descriptions of the intervention protocols, including session duration, intervention frequency, and total intervention period. The duration of VR interventions ranged from 4 to 12 weeks, with session lengths varying between 20 and 90 min. A summary of the study characteristics is presented in Table 3.

Quality assessment of the literature

A total of thirteen RCTs (Santos et al., 2019; Pullia et al., 2023; Liao, Yang & Wu, 2015; Goffredo et al., 2023; Pazzaglia et al., 2020; Gulcan et al., 2023; Maranesi et al., 2022; Kashif et al., 2022; Yang et al., 2016; Lichun & Rong, 2020; Rong, 2021; Feng et al., 2019; Xia, Jiang & Zheng, 2020) were included in this study. All studies met key methodological criteria, including random allocation, baseline similarity, intention-to-treat analysis, between-group statistical outcome analysis, and use of point measurements with difference values. Allocation concealment was reported in two studies, while eight studies reported blinding of outcome assessment. The Physiotherapy Evidence Database (PEDro) scores ranged from 6 to 8, with an average score of 6.77, indicating generally good methodological quality. No studies of low quality were identified. Detailed quality assessment results are presented in Table 4.

Table 3 Basic information of the included literature.

Inclusion of studies	Country	Sample size	Sex (m/f)	Age (years)	Course of disease	Modality	Duration of intervention	Frequency of intervention	Intervention period	Assessment tools	
		(T/C)	(T/C)	(T/C)	(T/C)	(T/C)					
Santos, P. 2019	Brazil	14/14	11/3	9/5	64.5 ± 9.8	66.6 ± 8.2	7.8+3.7 (year)	6.5+2.07 (year)	A+B/A	50 min/time	2 times a week	8 weeks	①②	
Pullia, M. 2023	Italy	10/10	13/7	65 ± 8.28	10.8 ± 7.75 (year)	B/A	20 min/time	4 times a week	5 weeks	④⑤	
Liao, Ying-Yi 2015	Taiwan	12/12	6/6	5/7	67.3 ± 7.1	64.6 ± 8.6	7.9 ± 2.7 (year)	6.4 ± 3.0 (year)	B+C/A	60 min/time	2 times a week	6 weeks	③⑤⑥	
Goffredo, Michela 2023	Italy	49/48	27/22	24/24	67.8 ± 6.6	67.8 ± 6.6	4.71 ± 4.58 (year)	5.27 ± 5.54 (year)	B/A	45 min/time	3–5 times a week	6–10 weeks	②④	
Pazzaglia, C. 2020	Italy	25/26	18/7	17/9	72 ± 7	70 ± 10	7.4 ± 7.7 (year)	4.8 ± 4.4 (year)	B/A	40 min/time	3 times a week	6 weeks	①	
Gulcan, K. 2023	Turkey	15/15	13/2	13/2	61.36 ± 7.36	59.27 ± 13.09	5.64 ± 7.36 (year)	5.27 ± 6.54 (year)	A+B/A	60 min/time	3 times a week	6 weeks	⑦	
Maranesi, E. 2022	Italy	16/14	6/10	9/5	72.7 ± 6.3	75.5 ± 5.4	–	–	A+B/A	50 min/time	2 times a week	5 weeks	⑥⑧	
Kashif, M. 2022	Pakistan	22/22	13/9	12/10	63.86 ± 4.57	62.32 ± 4.61	6.23 ± 1.85 (year)	6.55 ± 1.68 (year)	A+B/A	60 min/time	3 times a week	12 weeks	⑨	
Yang, Wen-Chieh 2016	Taiwan	11/12	4/7	5/7	72.5 ± 8.4	75.4 ± 6.3	9.4 ± 3.6 (year)	8.3 ± 4.1 (year)	B/A	50 min/time	2 times a week	6 weeks	①②	
Lichun Sun 2020	China	30/30	19/11	21/9	61.43 ± 7.34	62.54 ± 6.98	6.17 ± 2.39 (year)	6.44 ± 1.97 (year)	A+B/A	60 min/time	5 times a week	4 weeks	②⑤⑥⑩	
Rong Jang 2021	China	38/38	22/16	23/15	62.62 ± 3.75	61.92 ± 4.25	4.25 ± 0.84 (year)	4.31 ± 0.92 (year)	A+B/A	30 min/time	5 times a week	4 weeks	②⑥⑦⑩	
Hao Feng 2019	China	14/14	8/6	9/5	67.47 ± 4.79	66.93 ± 4.64	7.07 ± 1.44 (year)	6.60 ± 1.45 (year)	A+B/A	45 min/time	7 times a week	12 weeks	②③	
Min Xia 2020	China	15/15	11/4	12/3	65.99 ± 4.30	66.00 ± 8.55	–	–	A+B/A	30 min/time	3 times a week	4 times a week	②⑤	
Notes.

Note: Intervention modalities include: A –Routine Rehabilitation, B –Virtual Reality (VR) Training, and C –Treadmill Training. A dash (‘—’) indicates data not reported. Outcome measures are as follows: (①) DGI –Dynamic Gait Index; (②) TUG –Timed “Up & Go” test; (③) FGA –Functional Gait Assessment; (④) 6MWT –6-Minute Walking Test; (⑤) 10MWT –10-Meter Walking Test; (⑥) Step frequency (m/s); (⑦) Step length (cm); (⑧) POMA TOTAL –Tinetti’s Performance-Oriented Mobility Assessment; (⑨) UPDRS-III gait –Gait item from the Unified Parkinson’s Disease Rating Scale Part III; and (⑩) Step length asymmetry –Difference in step length between the right and left sides (Santos et al., 2019; Pullia et al., 2023; Liao, Yang & Wu, 2015; Goffredo et al., 2023; Pazzaglia et al., 2020; Gulcan et al., 2023; Maranesi et al., 2022; Kashif et al., 2022; Yang et al., 2016; Lichun & Rong, 2020; Rong, 2021; Feng et al., 2019; Xia, Jiang & Zheng, 2020).

Table 4 Literature quality assessment.

Inclusion of studies	A1	A2	A3	A4	A5	A6	A7	A8	A9	A10	Totals	
Santos, P. 2019	1	0	1	0	0	1	1	1	1	1	7	
Pullia, M. 2023	1	0	1	0	0	1	1	1	1	1	7	
Liao, Ying-Yi 2015	1	1	1	0	0	1	1	1	1	1	8	
Goffredo, Michela 2023	1	1	1	0	0	1	1	1	1	1	8	
Pazzaglia, C. 2020	1	0	1	0	0	0	1	1	1	1	6	
Gulcan, K. 2023	1	0	1	0	0	1	1	1	1	1	7	
Maranesi, E. 2022	1	0	1	0	0	0	1	1	1	1	6	
Kashif, M. 2022	1	0	1	0	0	1	1	1	1	1	7	
Yang, Wen-Chieh 2016	1	0	1	0	0	1	1	1	1	1	7	
Lichun Sun 2020	1	0	1	0	0	0	1	1	1	1	6	
Rong Jiang 2021	1	0	1	0	0	0	1	1	1	1	6	
Hao Feng 2019	1	0	1	0	0	1	1	1	1	1	7	
Min Xia 2020	1	0	1	0	0	0	1	1	1	1	6	
Notes.

PEDro Physiotherapy Evidence Database

A1 random assignment of participants to groups

A2 hidden assignment

A3 groups were similar at baseline

A4 all participants were blind

A5 all therapists were blind

A6 all assessors were blind

A7 measurement of at least one of the main outcomes was obtained from more than 85% of the participants

A8 intention-to-treat analysis was conducted

A9 results of statistical comparisons between groups for at least one main outcome were reported

A10 the study reported point and variability measures for at least one main outcome (Santos et al., 2019; Pullia et al., 2023; Liao, Yang & Wu, 2015; Goffredo et al., 2023; Pazzaglia et al., 2020; Gulcan et al., 2023; Maranesi et al., 2022; Kashif et al., 2022; Yang et al., 2016; Lichun & Rong, 2020; Rong, 2021; Feng et al., 2019; Xia, Jiang & Zheng, 2020)

Meta-analysis results

The meta-analysis demonstrated a statistically significant and clinically meaningful effect of VR-based training on gait performance in individuals with Parkinson’s disease. Pooled data from thirteen RCTs indicated that VR interventions led to moderate improvements in gait outcomes, with a SMD of 0.56 (95% Cl: 0.36 to 0.77, P <0.00001), as illustrated in Fig. 2. This effect size reflects a moderate clinical benefit and supports the potential of VR-based rehabilitation to enhance functional mobility and reduce gait-related disability in this population.

Figure 2 Combined total effect sizes for gait (Santos et al., 2019; Pullia et al., 2023; Liao, Yang & Wu, 2015; Goffredo et al., 2023; Pazzaglia et al., 2020; Gulcan et al., 2023; Maranesi et al., 2022; Kashif et al., 2022; Yang et al., 2016; Lichun & Rong, 2020; Rong, 2021; Feng et al., 2019; Xia, Jiang & Zheng, 2020).

Although the included studies varied in terms of VR modalities-most commonly non-immersive systems-and intervention parameters such as frequency, session duration, and total training period, the overall direction of effect remained consistently positive. These findings demonstrate the general applicability of VR-based rehabilitation across diverse clinical settings and provide a rationale for exploring how intervention characteristics influence treatment efficacy. To further investigate this, subgroup analyses were conducted based on intervention frequency, session duration, and total intervention period, as reported in the following section.

Subgroup analysis of moderating effects

To further investigate potential sources of heterogeneity in the effects of VR training on gait outcomes, subgroup analyses were conducted based on key moderator variables, as detailed in Table 5, which summarizes the corresponding effect sizes and confidence intervals. These variables included participant characteristics (age and disease duration) and intervention parameters (total intervention period, session length, and training frequency). For age, studies were stratified into two groups: ≤65 years and >65 years (Hindle, 2010; Cabreira & Massano, 2019). For disease duration, a cutoff of 8 years was used to form subgroups of ≤8 years and >8 years. Intervention duration was categorized as short-term (≤4 weeks) and long-term (>4 weeks); session length was classified as ≤30 min versus >30 min; and training frequency was divided into ≤3 sessions per week and >3 sessions per week. These subgroup analyses were performed to assess whether the effects of VR-based rehabilitation varied systematically across different demographic and intervention-related characteristics. The findings may help identify the optimal conditions for maximizing gait-related outcomes in individuals with Parkinson’s disease.

Table 5 Subgroup analysis of the effect of VR training on gait improvement in PD.

Outcome indicator	Number of studies included	I2/%	SMD (95% CI)	P-value	
Gait	13	66	0.56 (0.36, 0.77)	<0.00001	
Age	≤65 years old	8	51	0.48 (0.23, 0.74)	=0.00020	
	>65 years old	5	68	0.74 (0.46, 1.02)	<0.00001	
Course of disease	≤8 years	9	70	0.65 (0.38, 0.92)	<0.00001	
>8 years	2	0	0.23 (−0.20, 0.65)	=0.29000	
Intervention period	≤4 weeks	3	43	0.86 (0.63, 1.09)	<0.00001	
>4 weeks	10	62	0.46 (0.20, 0.72)	=0.00050	
Intervention Time	≤30 min/times	6	48	0.83 (0.62, 1.04)	<0.00001	
>30 min/times	7	58	0.37 (0.09, 0.65)	=0.01000	
Frequency of intervention	≤3 times	8	64	0.60 (0.28, 0.92)	=0.00030	
>3 times	5	67	0.64 (0.38, 0.89)	<0.00001	

Sensitivity analysis

To assess the robustness of the meta-analytic results and identify potential sources of heterogeneity, sensitivity analyses were conducted using a leave-one-out approach. Each included study was sequentially removed, and the pooled effect size was recalculated. The overall estimates remained stable throughout, indicating that no single study disproportionately influenced the results. However, substantial heterogeneity (I2 > 50%) persisted despite subgroup analyses. Although the sensitivity analysis did not isolate a single outlier, the variability may reflect underlying differences in study characteristics. In particular, heterogeneity could plausibly arise from variation in VR modalities (e.g., immersive vs. non-immersive systems), differences in baseline disease severity among participants, or inconsistencies in outcome measurement tools. These interacting factors were not uniformly reported across studies, limiting further stratified analyses. This limitation was taken into account during the interpretation of pooled estimates and the grading of evidence quality (Table 6).

Table 6 Combined effects of excluding individual study gait (Santos et al., 2019; Pullia et al., 2023; Liao, Yang & Wu, 2015; Goffredo et al., 2023; Pazzaglia et al., 2020; Gulcan et al., 2023; Maranesi et al., 2022; Kashif et al., 2022; Yang et al., 2016; Lichun & Rong, 2020; Rong, 2021; Feng et al., 2019; Xia, Jiang & Zheng, 2020).

Inclusion of studies	Effect size	95% CI	P	I2/%	
Santos, P. 2019	0.58	0.36, 0.79	<0.00001	68	
Pullia, M. 2023	0.60	0.40, 0.81	<0.00001	65	
Liao, Ying-Yi 2015	0.51	0.30, 0.72	<0.00001	67	
Goffredo, Michela 2023	0.61	0.40, 0.82	<0.00001	62	
Pazzaglia, C. 2020	0.57	0.36, 0.78	<0.00001	67	
Gulcan, K. 2023	0.61	0.41, 0.81	<0.00001	63	
Maranesi, E. 2022	0.60	0.39, 0.81	<0.00001	67	
Kashif, M. 2022	0.53	0.33, 0.73	<0.00001	65	
Yang, Wen-Chieh 2016	0.59	0.38, 0.80	<0.00001	68	
Lichun Sun. 2020	0.51	0.29, 0.74	<0.00010	67	
Rong Jiang. 2021	0.51	0.28, 0.73	<0.00001	65	
Feng, H. 2019	0.56	0.35, 0.77	<0.00001	67	
Min Xia. 2020	0.53	0.32, 0.74	<0.00001	67	

Publication bias

To evaluate the potential presence of publication bias, Egger’s test was performed for the primary outcome of gait function. The results indicated no significant publication bias (Egger’s test: p = 0.564, which exceeds the conventional threshold of 0.05). This suggests that the meta-analytic findings were unlikely to be influenced by selective publication of studies reporting positive results (Fig. 3).

Figure 3 Gait publication bias.

Evaluation of evidence quality

A total of thirteen randomized controlled trials were included in this review, yielding a mean PEDro score of 6.77, indicative of overall good methodological quality. Sensitivity analyses revealed that the pooled effect sizes remained consistent following the stepwise exclusion of individual studies, supporting the robustness and reliability of the findings. However, subgroup analyses suggested that heterogeneity across studies may be partially attributable to differences in disease duration, intervention frequency, and intervention duration. No significant publication bias was detected for gait-related outcomes, as indicated by Egger’s test. According to the GRADE framework (Fig. 4), the overall certainty of the evidence supporting the effect of VR-based interventions on gait improvement in individuals with Parkinson’s disease was rated as high (Liang et al., 2014).

Adverse events

None of the thirteen included randomized controlled trials reported any adverse events related to virtual reality training. This absence of reported complications suggests that VR-based interventions are generally safe and well tolerated by individuals with Parkinson’s disease. However, the lack of systematic adverse event monitoring in some studies warrants cautious interpretation and underscores the need for future trials to include standardized safety assessments.

Figure 4 GRADE quality of evidence evaluation.

Discussion

This meta-analysis confirmed that VR training significantly improves gait function in individuals with PD, consistent with previous systematic reviews (Ler, Sunzi & Dai, 2019; Chen et al., 2020). The pooled SMD for gait outcomes was 0.56, which corresponds to a moderate clinical effect based on established interpretation thresholds. This suggests that VR-based interventions are not only statistically significant but also associated with noticeable improvements in functional mobility. Although the minimal clinically important difference (MCID) for gait-related outcomes in PD remains under discussion, a threshold of SMD 0.5–0.6 has been proposed to indicate meaningful functional improvement, reinforcing the clinical relevance of our findings.

Recent evidence indicates that combining VR with treadmill training can improve motor and cognitive outcomes in individuals with PD by engaging sustained attentional control and motor learning mechanisms, activating cerebellar pathways to compensate for basal ganglia dysfunction, and enhancing top-down networks involved in motor–cognitive integration (Ruidan, Peiling & Weijun, 2018; Liao, Yang & Wu, 2015; Mirelman, Maidan & Herman, 2011). Building on these neural mechanisms, short-duration and high-frequency VR protocols may offer superior benefits by leveraging residual neuroplasticity in patients with PD. Repetitive, task-specific training has been shown to induce neural adaptation, support the reorganization of motor circuits, and improve gait stability and coordination (Laver, George & Thomas, 2011; Yogev-Seligmann, Hausdorff & Giladi, 2010). Moreover, limiting each session to ≤30 min may better align with the adaptive capacity of the PD nervous system, facilitating greater cortical engagement and inter-regional connectivity during motor execution (Kleim & Jones, 2008). These mechanisms may underlie the rapid acquisition and retention of motor patterns frequently observed in immersive VR environments. Crucially, early implementation of VR-based rehabilitation may maximize motor recovery by capitalizing on preserved neurocognitive resources before irreversible neurodegeneration sets in Tanner (1996).

Subgroup analyses further revealed that VR training was particularly effective in individuals with a disease duration of ≤8 years, suggesting greater responsiveness in the earlier stages of PD. This finding supports the hypothesis that early application of VR rehabilitation can maximize residual neuroplasticity by targeting the therapeutic window before the progression of neurodegenerative damage (Canning et al., 2020; De Natale et al., 2025). Moreover, the most effective protocols were characterized by session durations ≤30 min, frequencies of ≥2 times per week, and total intervention durations ≤4 weeks. These short-term, high-frequency regimens may optimize motor adaptation while minimizing the fatigue or cognitive burden commonly observed in PD patients (Wu et al., 2022). While these findings are promising, they should be interpreted with caution within the broader research context, as previous reviews have generally supported the potential of VR training to improve various gait-related parameters in Parkinson’s disease, though variability in outcome measures warrants careful consideration; however, few have explored whether intervention effects differ across patient subgroups or training parameters. In particular, critical variables such as training frequency, session duration, and intervention length are often not systematically stratified, limiting the identification of optimized intervention protocols. Our subgroup analysis provides preliminary evidence that both the timing of intervention and the structure of VR training may influence therapeutic outcomes. This aligns with current neurorehabilitation frameworks emphasizing individualized adaptation and early intervention strategies, which may guide the development of more tailored clinical applications (Pelosin et al., 2021; Goffredo et al., 2023). At the same time, considerable variability was observed across studies in terms of VR type (immersive vs. non-immersive), feedback mechanisms (visual, auditory, haptic), and task complexity. This structural heterogeneity highlights the absence of standardized protocols in VR rehabilitation for PD. Such inconsistency not only limits the comparability of findings across studies but also poses challenges for implementation and generalization of VR-based therapies. Future trials should aim to establish greater standardization in the design and reporting of VR interventions to enhance their replicability and clinical applicability (Ren, Wang & Liu, 2024).

Clinical Implications

The results of this meta-analysis provide important insights for clinical practice. The overall effect size of VR-based training on various gait-related parameters in patients with Parkinson’s disease was moderate (SMD = 0.56). Since the analysis included multiple gait outcome measures rather than a single uniform endpoint, interpretations of “gait improvement” should be approached with caution. Although the intervention demonstrated statistical significance, its clinical relevance may vary depending on specific gait parameters and patient characteristics. Subgroup analyses indicated a trend toward better responsiveness to VR training in patients with disease duration of ≤8 years (SMD = 0.65), suggesting that early-stage interventions may better leverage neuroplasticity to enhance rehabilitation outcomes; however, further research is needed to clarify the effects across different gait domains.

Moreover, intervention protocols with session durations of ≤30 min and total intervention periods of ≤4 weeks demonstrated the largest effect sizes (SMD = 0.83 and 0.86, respectively). These findings support the feasibility and clinical acceptability of short-duration, high-frequency, task-oriented training programs, which may improve adherence while minimizing fatigue. Based on these results, VR-based gait rehabilitation is recommended as an individualized, non-pharmacological intervention, particularly for patients in the early stages of Parkinson’s disease. When implemented with appropriately tailored parameters, VR may serve as an effective adjunct to traditional rehabilitation strategies to enhance functional gait recovery.

Limitation

One important limitation of this study is that outcome assessment in many of the included trials was not blinded. As gait performance was often measured using subjective or semi-subjective tools (e.g., TUG, DGI, UPDRS gait subscore), the absence of assessor blinding may have introduced performance or detection bias, potentially leading to an overestimation of intervention effects. This methodological weakness was prevalent across multiple studies and warrants caution when interpreting the clinical implications of the findings. Although we evaluated study quality using the PEDro scale and rated the overall evidence as high according to the GRADE framework, the potential risk of bias due to unblinded assessments remains a critical concern.

Another important limitation is that we did not perform a detailed classification of VR intervention modalities (e.g., immersive vs. non-immersive, game-based vs. task-oriented training). This decision was made due to the limited number of eligible studies and considerable heterogeneity in VR system design, content, and implementation procedures across trials. While this approach preserved the overall sample size and statistical power, it may have obscured modality-specific effects and contributed to unexplained heterogeneity. Future studies with more standardized and well-described VR protocols are needed to enable more granular subgroup analyses.

Conclusion

This systematic review and meta-analysis demonstrates that VR-based training significantly improves gait function in individuals with Parkinson’s disease. The therapeutic effects appear to be moderated by disease duration and exhibit a dose–response relationship, particularly with respect to the total intervention period and the duration of individual sessions. In contrast, patient age and training frequency showed no consistent influence on outcomes, potentially due to the compensatory role of neuroplasticity, individual adaptability to immersive technologies, and the cognitive engagement elicited by VR environments. These findings offer evidence-based support for incorporating VR training into personalized rehabilitation programs for Parkinson’s disease. Nevertheless, the current body of research remains limited in terms of the diversity of VR modalities and the characterization of patient subgroups. Future high-quality trials are warranted to examine the comparative effectiveness of different VR systems and to optimize intervention protocols based on disease stage, cognitive status, and patient preferences.

Supplemental Information

Supplemental Information 1 PRISMA checklist

Supplemental Information 2 Rationale and Contribution Statement

Additional Information and Declarations

Competing Interests

Author Contributions

Data Availability

The authors declare there are no competing interests.

Shanan Yu conceived and designed the experiments, analyzed the data, authored or reviewed drafts of the article, and approved the final draft.

Yu Zhu performed the experiments, prepared figures and/or tables, and approved the final draft.

Yanfei Yang analyzed the data, authored or reviewed drafts of the article, and approved the final draft.

The following information was supplied regarding data availability:

Raw data was not generated in this systematic review.

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
