# Peer review of "Effects of virtual reality intervention dosage on gait performance in Parkinson’s disease: a systematic review and meta-analysis"

_PeerJ, doi:10.7717/peerj.20320_

## Round 0.1 · original submission · Major Revisions

· Academic Editor

Major Revisions

**Language Note:** The review process has identified that the English language must be improved. PeerJ can provide language editing services - please contact us at [email protected] for pricing (be sure to provide your manuscript number and title). Alternatively, you should make your own arrangements to improve the language quality and provide details in your response letter. – PeerJ Staff

Reviewer 1 ·

Basic reporting

The English language is generally understandable and professionally written, although there are occasional awkward phrasings and formatting errors (e.g., "f8 years" instead of "<8 years").
The abstract could better define key terms like SMD and provide more interpretation (e.g., explain what an SMD of 0.56 means in clinical terms).
References within the text sometimes contain incorrect or placeholder citations (e.g., "[Error! Reference source not found.]") which need correction for traceability.

Experimental design

The authors mention a "double-blind" literature screening process, which is not applicable in this context; a clarification on what is meant (independent review by two authors) would suffice.

More detail on how VR interventions differed across studies (immersive vs. non-immersive, game-based vs. training modules) would enhance interpretation.

The exclusion criteria should explicitly mention whether studies combining VR with non-rehabilitation therapies (pharmacological) were excluded.

Validity of the findings

High heterogeneity (I² > 50%) persists despite subgroup analysis. While this is acknowledged, further exploration of heterogeneity sources (VR modality, baseline severity) is warranted.

The lack of blinding in many included studies could introduce bias, particularly for subjective gait assessments; this is a critical limitation that should be more explicitly discussed in the conclusion.

The analysis would benefit from a discussion of minimal clinically important differences (MCIDs) to contextualize statistical significance in clinical terms.

Additional comments

The study is methodologically sound and clinically relevant but needs technical corrections, clarification of minor methodological points, and strengthened discussion of clinical applicability.

Reviewer 2 ·

Basic reporting

English language and grammar revision is strongly recommended to ensure clarity and academic tone throughout.

The Introduction does not adequately establish the background, significance, or rationale for the review. The purpose of the study is not clearly articulated in the abstract and introduction. A clear and focused objective is essential for guiding the reader and establishing the context of the review and analysis.

The Discussion fails to provide critical analysis of the findings in the context of existing literature.

Experimental design

There are inconsistencies in participant numbers extracted from the included studies—reported as 525 in some instances and 541 in others—which raises concerns about the accuracy and reliability of data synthesis.

The inclusion and exclusion criteria appear contradictory and inconsistently applied:
Although the inclusion criteria specify participants "aged between 60 and 80 years," some studies included participants younger than 60 years, which violates the stated criteria.
The inclusion criteria mention that the experimental group received "VR training either as a standalone intervention or in combination with conventional rehabilitation." However, the exclusion criteria state that studies using "combined interventions in the experimental group (e.g., VR combined with acupuncture or other therapies)" were excluded. This contradiction requires clarification and a consistent approach to study selection.

Validity of the findings

Given the issues listed above—particularly the inconsistencies in data reporting, lack of methodological rigor, and insufficient clarity in key sections—I recommend rejection of this manuscript in its current form. The study may have potential if substantially revised with clearer objectives and consistent criteria, but as it stands, it does not meet the standards required for publication.

Reviewer 3 ·

Basic reporting

The article follows a basic academic structure, refinement is needed for clarity, and accuracy.
Many tables are displaying errors and are not correctly formatted, making them difficult to read and interpret.

Experimental design

1. Title of the study
The title should be concise and descriptive. Currently, there is a mismatch between the title and the stated objective. One emphasises gait improvement, while the other focuses on intervention dosage—this inconsistency needs to be resolved.

2. Objective:
The objective is repeated and lacks specificity. It should clearly define what the study aims to assess—e.g., the effect of VR intervention dosage on gait performance.

3. Introduction
The introduction lacks focus and clarity. Key terms like NIVR, EEG, and gait variables are mentioned without adequate explanation. Additionally, the rationale linking the intervention (VR) to gait outcomes is underdeveloped. A major revision is needed to align the background with the title and objectives.

4. Methodology/Procedure:

The PICO table is unclear and should be revised for consistency and completeness.
There is a contradiction between line 137 and line 149 regarding the inclusion/exclusion of VR combined with acupuncture or other therapies.
“Other therapies” in the exclusion criteria should be clearly defined.

The primary outcome is stated as gait function, but the parameters used (e.g., step length, gait speed, cadence, etc.) are not reported or linked to the respective outcome tools.

5. Results/Findings:

The results section lacks basic and essential reporting, such as summary tables or narrative explanations of key outcomes.
There is no detailed explanation of subgroup analyses within the meta-analysis, especially regarding different VR types or dosage effects.
Findings should be supported with descriptive statistics and forest plots where applicable.

Validity of the findings

The discussion does not explain the observed improvements in gait performance.
There is no clear explanation for why VR may be effective(eg; types of VR which parameteres are improved etc).
I think references to support claims in the discussion are insufficient or missing.
The discussion should be significantly improved by linking results to previous literature and providing clinical or theoretical implications.

Additional comments

nil

Annotated reviews are not available for download in order to protect the identity of reviewers who chose to remain anonymous.

---

## Round 0.2 · accepted · Accept

· Academic Editor

Accept

Thank you for revising your manuscript to address the concerns of the reviewers. Reviewer 2 now recommends acceptance and I am satisfied that the comments of reviewers 1 and 3 have been addressed. The manuscript is now ready for publication.

Reviewer 2 ·

Basic reporting

Thank you for addressing all concerns and comments.

Experimental design

Thank you for addressing all concerns and comments.

Validity of the findings

Thank you for addressing all concerns and comments.

Additional comments

Thank you for addressing all concerns and comments.